



# Physics-motivated Cell-octree Adaptive Mesh Refinement in the Vlasiator 5.3 Global Hybrid-Vlasov Code

Leo Kotipalo[1], Markus Battarbee[1], Yann Pfau-Kempf[1], and Minna Palmroth[1,2]

[1]University of Helsinki
[2]Finnish Meteorological Institute

**Correspondence:** Leo Kotipalo (leo.kotipalo@helsinki.fi)

**Abstract.** Automatically adaptive grid resolution is a common way of improving simulation accuracy while keeping the computational efficiency at a manageable level. In space physics adaptive grid strategies are especially useful as simulation volumes are extreme, while the most accurate physical description is based on electron dynamics and hence requires very small grid cells and time steps. Therefore, many past global simulations encompassing e.g. the near-Earth space have made tradeoffs in

terms of the physical description and used laws of magnetohydrodynamics (MHD) that require less accurate grid resolutions. Recently, using supercomputers, it has become possible to model the near-Earth space domain with an ion-hybrid scheme going beyond the MHD-based fluid dynamics. These simulations, however, must develop a new adaptive mesh strategy beyond what is used in MHD simulations.

We developed an automatically adaptive grid refinement strategy for ion-hybrid Vlasov schemes, and implemented it within

the Vlasiator global solar wind - magnetosphere - ionosphere simulation Vlasiator. This method automatically adapts the resolution of the Vlasiator grid using two indices: one formed as a maximum of dimensionless gradients measuring the rate of spatial change in selected variables, and the other derived from the ratio of the current density to the magnetic field density perpendicular to the current. Both these indices can be tuned independently to reach a desired level of refinement and computational load. We test the indices independently and compare the results to a control run using static refinement.

The results show that adaptive refinement highlights relevant regions of the simulation domain and keeps the computational effort at a manageable level. We find that the refinement shows some overhead in rate of cells solved per second. This overhead can be large compared to the control run without adaptive refinement, possibly due to resource utilisation, grid complexity and issues in load balancing. These issues lay a development roadmap for future optimisations.



## 1 Introduction

Due to the practical difficulty of gathering in-situ measurements, simulations are an indispensable tool for space physics research. The two primary families of models are kinetic models where plasma is described as a collection of particles with position and velocity, and fluid models where particle species are simplified to a fluid with macroscopic spatial properties. Hybrid methods combine these two approaches, typically modeling ions kinetically and the much lighter electrons as a fluid. Particle-in-cell (PIC) is a notable kinetic method, simplifying large numbers of particles into macroparticles with a single

position and velocity (Nishikawa et al., 2021). Another way to describe the particles is through a six-dimensional distribution function $f(\mathbf{x}, \mathbf{v})$ describing particle density in position and velocity space. The distribution is evolved in time according to the Vlasov equation and this method is thus called the Vlasov method (Palmroth et al., 2018). A commonly used fluid method is magnetohydrodynamics (MHD), where all particle species are simplified into a single fluid (Janhunen et al., 2012).

Vlasiator is a hybrid-Vlasov plasma simulation that models ions kinetically and electrons as a massless, charge-neutralizing

fluid, used for global simulation of near-Earth plasma. Time-evolution of the distribution is semi-Lagrangian: the function's value is stored on a six-dimensional Eulerian grid of three Cartesian spatial and velocity dimensions, sampled and transported along their characteristics via Lagrange's method, and then sampled back. This is done one dimension at a time using the SLICE-3D algorithm (Zerroukat and Allen, 2012). Electron charge density everywhere is taken to be equal to the ion charge density. Magnetic fields are solved using Faraday's law and electric fields using Hall MHD Ohm's law and Ampère's law

with the displacement current neglected, added to a static background dipole field approximating the geomagnetic dipole. The simulation uses OpenMP for threading and MPI for task parallelism, with load balancing handled via the Zoltan library (Devine et al., 2002). A more in-depth description of the model used is provided by von Alfthan et al. (2014) and Palmroth et al. (2018).

As a global plasma simulation, Vlasiator's problem domain encompasses the entire magnetosphere and enough of its surroundings to model interactions with the solar wind. This domain ranges from shock interfaces with discontinuous conditions

to areas of relative spatial homogeneity. Due to this diversity, using a static, homogeneous grid for calculations is suboptimal. This paper explores the automatic local adjustment of spatial resolution using a method called adaptive mesh refinement or AMR (Berger and Jameson, 1985), examining its performance impact and assessing whether AMR enhances the simulation results.

The paper is organised as follows: Section 1 describes the problem domain and adaptive mesh refinement. Section 2 describes

methods used in Vlasiator and the implementation of AMR. Section 3 examines the qualitative effect on the simulation grid and quantitative performance impact. Section 4 summarizes the findings and discusses further avenues of development.

### 1.1 Problem domain

With three spatial and velocity dimensions, a uniform grid resolving the relevant kinetic scales can become unreasonably demanding to calculate. For simulation accuracy in the magnetospheric domain, some of the surroundings around the inner

magnetosphere and magnetotail need to be resolved; otherwise phenomena such as magnetic reconnection and magnetosheath waves (Dubart et al., 2020) cannot be described with sufficient accuracy. Using Earth radii ($R_E = 6.371 \times 10^6 \, \mathrm{m}$), take for



example a box sized $120\,\mathrm{R_E}$ in each dimension with $|y|, |z| \leq 60\,\mathrm{R_E}$ and $x \in [-100; 20]\,\mathrm{R_E}$ with $\Delta x = 1000\,\mathrm{km}$ for resolving kinetic effects while maintaining reasonable computational cost for a total of $4.5 \times 10^8$ spatial cells. Then resolve typical velocities with $|v_i| \leq 4000\,\mathrm{km\,s^{-1}}$ to match observed velocities and $\Delta v = 40\,\mathrm{km\,s^{-1}}$ to resolve kinetic effects resulting in

$8 \times 10^6$ velocity space cells per spatial cell (Pfau-Kempf et al., 2018). This results in $3.6 \times 10^{15}$ phase space cells taking about $14\,\mathrm{PiB}$ of memory stored as single-precision floating point numbers (Ganse et al., 2023), too much for any current supercomputer to handle. Grid dimensions here are sourced from Ganse et al. (2023); however, the memory figure differs due to a calculation error in that article.

Modeling the velocity space allows for representation of kinetic effects such as certain wave modes (Kempf et al., 2013;

Dubart et al., 2020) and instabilities (von Alfthan et al., 2014; Hoilijoki et al., 2016). However, this increases the computational load considerably compared to MHD methods where each spatial cell only stores a few moments instead of the entire velocity space. In a fully kinetic model the overhead would be even greater as this would involve resolving the phase space up to the kinetic scales of electrons. This would require shrinking spatial cells and shortening the timestep by a factor proportional to the mass ratio between protons and electrons of around $1\,836$.

Representing the velocity space sparsely can be used to alleviate the problem of dimensionality. Phase-space density is very low in most of the velocity space, so cells can be pruned without significant impact on simulation results. By limiting the velocity space to cells that pass a minimum density threshold, savings in computational load of up to two orders of magnitude can be achieved (von Alfthan et al., 2014); with $f_{\mathrm{min}} = 10^{-15}\,\mathrm{s^3 m^{-6}}$ memory savings of $98\,\%$ can be achieved with a mass loss of less than a percent (Pfau-Kempf et al., 2018). This allows for 2D simulations with three velocity space dimensions, but

adding a third spatial dimension requires spatial optimizations.

## 1.2 Adaptive Mesh Refinement

Due to the global nature of Vlasiator, the required resolution of simulation varies greatly between different regions of the magnetosphere illustrated in Figure 1. Particularly at shock surfaces and current sheets, the properties of plasma change rapidly over a short distance, requiring high spatial resolution. Currently Vlasiator models the solar wind as a constant Maxwellian

inflow, making the upstream solar wind homogeneous. Upstream plasma phenomena on a kinetic scale are beyond the scope of the simulation, as modeling them using Vlasov methods is unfeasible on a global scale due to the phase-space requirements outlined in the previous section.

To optimize simulation of a nonhomogenous problem, the spatial grid itself can have variable resolution. Regions of high interest and large spatial gradients can be modeled in a higher resolution than other areas. If the problem domain is well

known, this can be statically parametrized such that the grid is the same from start to finish. Alternatively, refinement can be done dynamically during runtime based on simulation data; this is called adaptive mesh refinement, or AMR.

AMR can be implemented in a block-based manner refining rectangular regions such as in the AMReX framework (Zhang et al., 2021), the hybrid-PIC simulation A.I.K.E.F. (Müller et al., 2011) and the MHD code BATS-R-US used in SWMF (Gombosi et al., 2021) and MHD-AEPIC (Wang et al., 2022), or cell by cell as in the grid library dccrg (Honkonen et al., 2013)

used in the MHD simulation GUMICS (Janhunen et al., 2012) and in Vlasiator. Block-based AMR provides easier optimization





and communication as each block is a simple cartesian grid and interfaces between refinement regions are minimized, but this limits the granularity of refinement, as refining entire blocks may create an excessive amount of refined cells (Stout et al., 1997).

This paper focuses on the cell-based approach, where the local value of a *refinement index* calculated from the cell data determines the refinement of the cell. Each cell has a refinement level with 0 being the coarsest; refining it splits the cell into smaller *children* on a higher refinement level. Each cell has a unique parent; unrefining or coarsening a cell merges it with its *siblings*, children of the same parent, back to the parent cell. Generally refinement in such a scheme may have an arbitrary shape. Cell-based refinement is sometimes called *quadtree* or *octree* refinement when splitting cubic cells into four or eight equal children in two or three dimensions respectively.





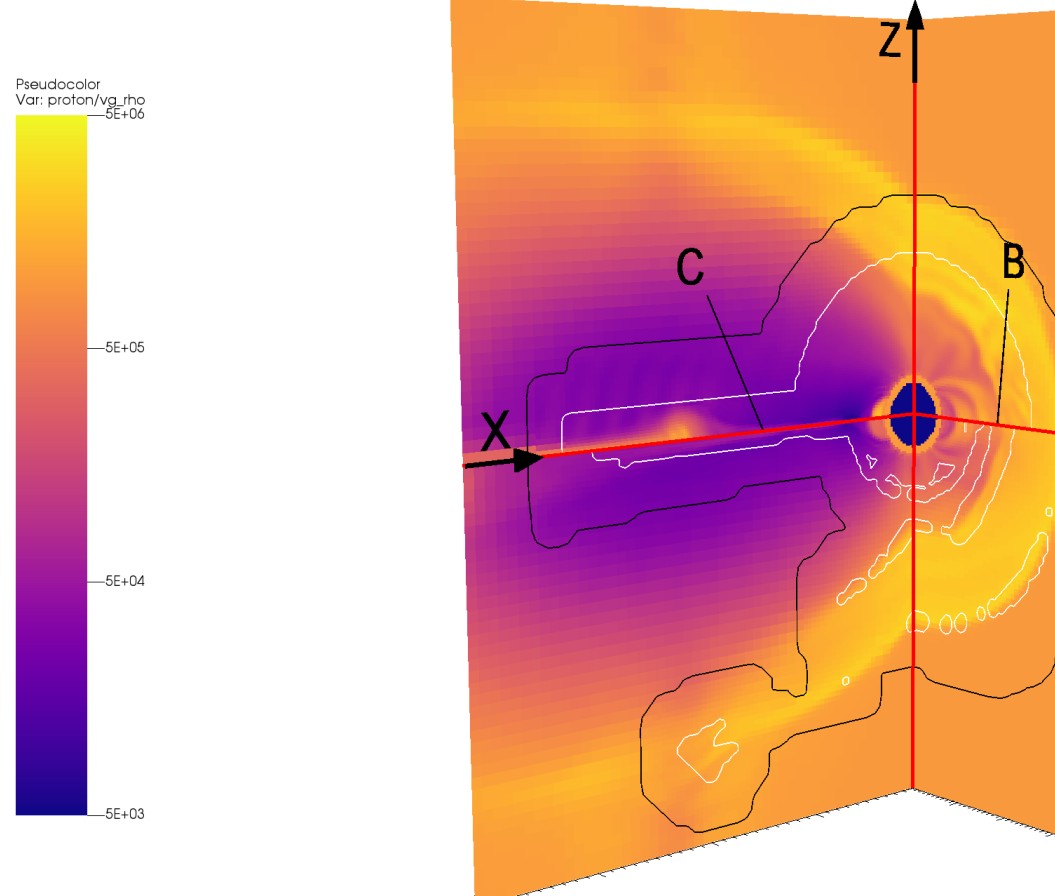

**Figure 1.** Overview of the global magnetospheric domain, tail side $XZ$- and $YZ$-planes in GSE coordinates, with $x$ positive sunward and $z$ northward perpendicular to the ecliptic plane. A is the bow shock, B the magnetopause and C the tail current sheet; these three regions are of particular interest, with variables changing rapidly over a short distance. The variable plotted is proton particle density, with two levels of refinement contours on top: the inside of the region outlined in black is higher resolution than the outside and the inside of the white outlines is the highest resolution. Two runs are plotted here, separated by the $XY$-plane marked by the red horizontal lines. The north side (above) is from a control run using static refinement, and the south side (below) from a run using adaptive refinement.



## 2  Methods

### 2.1  Spatial refinement

The sparse velocity space described in Section 1.1 is sufficient for hybrid simulations in two spatial dimensions, but three-dimensional simulations require additional spatial optimizations. The method used is cell-based spatial refinement, implemented in Vlasiator 5 (Pfau-Kempf et al., 2024) in a static manner by parametrizing regions to simulate at a finer spatial resolution (Ganse et al., 2023). The adaptive grid is provided by a library called distributed cartesian cell-refined grid or dccrg (Honkonen, 2023), which also communicates data between processes and provides an interface for the load balancing library Zoltan (Devine et al., 2002). Each cell keeps track of the processes cells in its neighborhood belong to, allowing remote communication.

Initially, each cubic cell starts at refinement level 0. These cells are refined by splitting them into eight equally sized cubic children, i.e. splitting in half along each cartesian direction. The children of a cell have a refinement level one higher than their parent. Dccrg cells have a neighborhood defined by a distance of the cell's own size for ghost data. Vlasiator's semi-Lagrangian solver has a stencil width of two cells, and a neighborhood of three cells is used to catch all edge cases. Neighbors are required to be at most one refinement level apart, so a cell of level 0 has a minimum of six level 1 cells between itself and a level 2 cell (Honkonen et al., 2013). The grid also has a maximum refinement level given as a parameter.

Static refinement is configured to refine a sphere around the inner boundary up to level 2, and the tail box and the nose cap up to level 3. This is done when starting a simulation run, and the refinement remains constant throughout. The spherical refinement and nose cap are meant to catch the magnetopause, and the tail box the magnetotail current sheet. The top half of Figure 1 shows this static refinement, with the two slices shown in full in Figure 2. The performance gains of this method are demonstrated by Ganse et al. (2023).

Note that spatial refinement is only applied to the 3D spatial grid containing the distribution function. Updating the electromagnetic field is relatively cheap so the field solver grid is homogeneous, matching the maximum refinement level of the Vlasov grid. After each Vlasov solver timestep moments of the distribution function are copied over to the field solver grid, to multiple field solver cells in case of resolution mismatch. These moments are used to evolve the field, and after the field solver timestep fields are copied from the field solver grid to the Vlasov grid, taking an average if necessary. To prevent sampling artifacts, a low-pass filter is used when the local resolutions don't match (Papadakis et al., 2022).

### 2.2  Shortcomings of static refinement

Examining Figure 2 reveals some issues with static refinement. Note that the parametrized spherical region does not follow the shape of the shock exactly; in this case an elliptic shape would refine less of the spatially homogeneous solar wind and catch more shock dynamics.

We may also consider situations where the refinement parameters fit poorly. Solar wind is not static, so under varying conditions the regions of interest may shift. Refinement regions are symmetric with respect to the $XY$ and $XZ$ planes, so they don't perfectly fit structures for an oblique solar wind or tilted geomagnetic dipole field either. Reparametrization is not too







**Figure 2.** Contour plot of static refinement on particle density, two slices. Note that the spherical refinement does not follow the shape of the shock; the second refinement level extends outside it and is circular rather than an arc.





difficult using trial and error, but it's unnecessary work for the end-user. In a dynamic simulation we might even find these regions shifting during a single run, making a static refinement from start to finish necessarily suboptimal.

A solution to these issues is to use adaptive mesh refinement for dynamic runtime refinement. With properly chosen parameters refinement should be better optimized, easier to tune and more adaptive to changing conditions. Re-refining with sufficient frequency also allows following dynamic structures. Quantitative comparisons between the refinement methods are given in Section 3.

## 2.3    Refinement indices

We first introduce the refinement index $\alpha_1$, a maximum of dimensionless gradients based on the index used in GUMICS (Janhunen et al., 2012):

$$\alpha_1 = \max \begin{cases} \frac{|\Delta \rho|}{\hat{\rho}} & \text{(a)} \\ \frac{|\Delta U|}{\widehat{U}} & \text{(b)} \\ \frac{(\Delta \mathbf{p})^2}{2\rho \widehat{U}} & \text{(c)} \\ \frac{(\Delta \mathbf{B})^2}{2\mu_0 \widehat{U}} & \text{(d)} \\ \frac{|\Delta \mathbf{B}|}{\widehat{B}}. & \text{(e)} \end{cases} \tag{1}$$

These gradients are of particle density (a), total (b), plasma (c) and magnetic field energy (d) density scaled to total energy density, and magnetic flux density (e). The contribution of the electric field energy density is considered negligible:

$$U \approx \frac{\mathbf{p}^2}{2\rho} + \frac{\mathbf{B}^2}{2\mu_0}. \tag{2}$$

In GUMICS, the magnetic field of the Earth's dipole is removed from $\mathbf{B}$, leaving the perturbed field $\mathbf{B}_1$ in the determination of these ratios. In Vlasiator we find that better refinement results from using the full magnetic field in (1). The simulation is given a refinement threshold and a coarsening threshold as parameters; for example a cell might be coarsened if $\alpha_1 < 0.3$ and refined if $\alpha_1 > 0.6$.

The way this works is each cell is compared pairwise to all cells that share a face with it, with $\Delta a$ being the difference and $\hat{a}$ the maximum in quantity $a$ between those two cells. The maximum of all these comparisons is the final value of $\alpha_1$.

     Plots of the constituents of $\alpha_1$ in the $XZ$-plane near the $x$-axis are given in Figure 3. Comparing this to Figure 2, the magnetotail and dayside magnetopause are clearly visible along with the front of the shock, but the inner magnetosphere is less highlighted.

A second refinement index is derived from current density and the magnetic field:

$$\alpha_2 = \frac{\mu_0 J}{B_\perp + \epsilon} \Delta x, \tag{3}$$

where $\mathbf{J} = \frac{1}{\mu_0} \nabla \times \mathbf{B}$ is the current density in the cell, $B_\perp$ the density of the magnetic field perpendicular to the current, $\epsilon$ a small constant to avoid dividing by zero and $\Delta x$ the edge length of the cell. This is used to detect magnetic reconnection





regions in the MHD-AEPIC model (Wang et al., 2022) in order to embed PIC regions, which is similar in aim to the spatial refinement sought in this work. Consider that $\alpha_2$ is dimensionless: $B_\perp / J$ gives a characteristic length scale, which is compared to $\Delta x$. We can again use a refinement and coarsening threshold as with $\alpha_1$.

As the framework for adaptive runtime refinement is implemented, developing new refinement indices is simple. So far, the Larmor radius $r_\mathrm{L}$ and the ion inertial length $d_\mathrm{i}$ have been evaluated but deemed to be less usable by themselves than the current indices. Combining them to an aggregate index in a similar way to $\alpha_1$ could prove useful and will be the topic of further investigations.

## 2.4 Interpretation of refinement thresholds

Refinement thresholds have a physical meaning as the maximum allowed gradient or perpendicular current in a cell before it's refined. A useful formulation would be to consider the *target* refinement level that a cell would be refined towards. $\alpha_2$ has an explicit dependency on $\Delta x$ while the constituents of $\alpha_1$ can generally be written as $\frac{\nabla y}{\hat{y}} \Delta x$. Taking a general refinement parameter $\alpha := y\Delta x$, its refinement criterion for a threshold $b$ is:

$$\alpha := y\Delta x > b$$
$$y\Delta x_0 \cdot 2^{-r} > b$$
$$y\Delta x_0 > b \cdot 2^r$$
$$\log_2\left(y\Delta x_0\right) > \log_2 b + r$$
$$\alpha' := \log_2\left(y\Delta x_0\right) - \log_2 b > r, \tag{4}$$

with the substitution $\Delta x = \Delta x_0 \cdot 2^{-r}$ using the zeroth level cell size $\Delta x_0$ and refinement level $r$. Thus, we can define $\alpha'$ as the target refinement level of the cell.

With his rescaling we now have a modified index where a cell's target refinement level is *at least $\alpha'$*, which can be calculated straightforwardly using the original index $\alpha$ and its refinement threshold $b$. This also gives the natural choice of the unrefinement threshold as $b/2$. In practice, a cell would be refined if $\alpha' > r$ and unrefined if $\alpha' < r - 1$, with $\log_2 b$ as a shift parameter.

The logical meaning of negative values is that the ideal cell size in a region is larger than the coarsest level of refinement in the grid. In practice increasing the size of the coarsest cells isn't necessary, as these regions typically have the least velocity space cells and contribute little to the overall computational load. A very large $\Delta x_0$ would also cause issues with induced refinement. In the configuration used in this work, a cell of level $-7$ would be larger than the entire grid, while having a single cell of level $-5$ would limit the maximum refinement found anywhere within the domain to $-3$.

Plots of $\alpha_1'$ and $\alpha_2'$ are given in Figure 4. Comparing this to Figure 2, the indices are more narrowly localized to the tail current sheet, shock and magnetopause than the static refinement. Additionally, a foreshock at $x \approx -30\,\mathrm{R_E}$ is picked up by both indices; refining such moving structures is impossible using static refinement. The two indices are somewhat similar but distinct; particularly the sub-solar shock is resolved better by $\alpha_1$, as $\alpha_2$ focuses on detecting current sheets which do not occur on the shock surface.



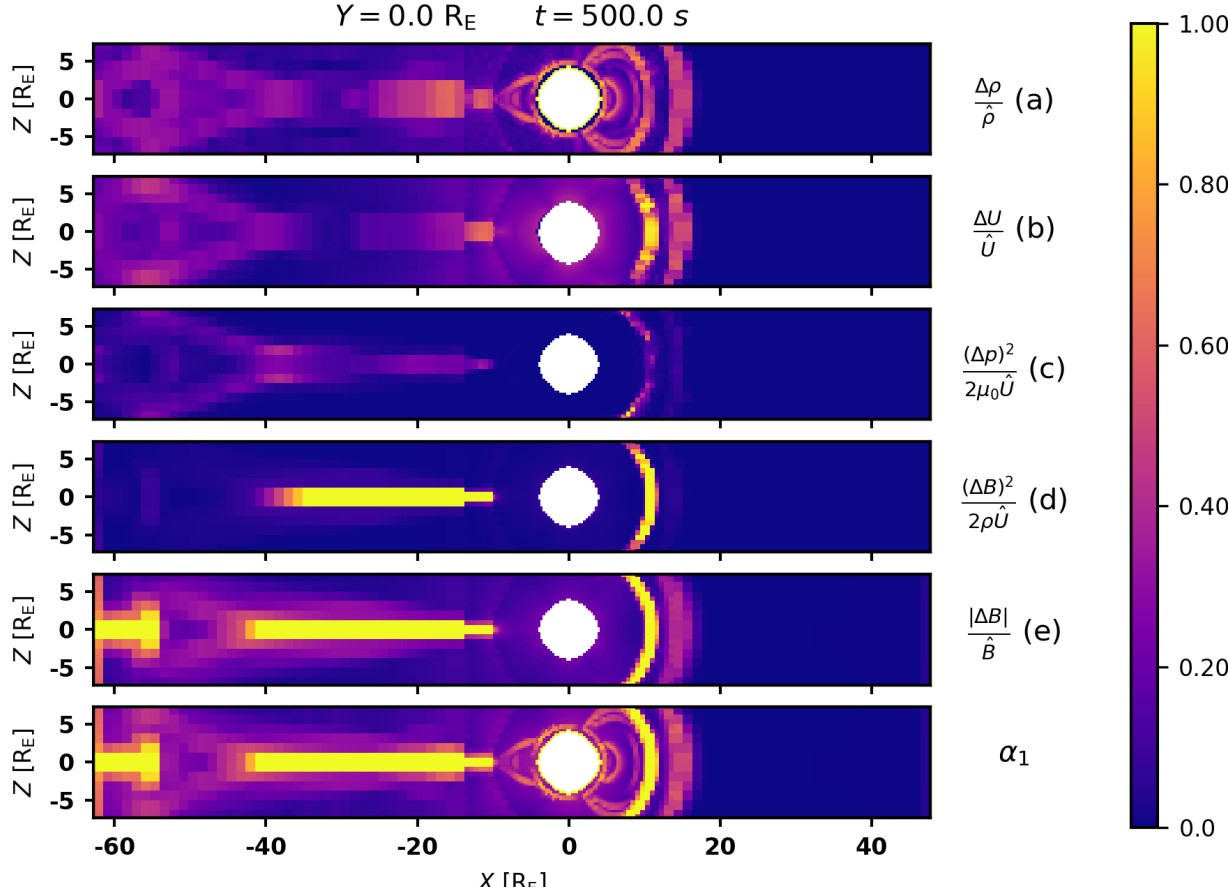

**Figure 3.** Plots of the constituent gradients of $\alpha_1$ near the $X$-axis at $y = 0$, and their maximum $\alpha_1$. Letters correspond to Equation (1). Gradients are clipped to a maximum of 1, as this is the maximum of (a) and (b). Particle density (a) and total energy density (b) seem to discern the magnetopause and shock, while the field energy density (d) and magnetic flux density (e) highlight the magnetopause and tail current sheets. Kinetic energy density (c) seems to have a minor effect on the value of $\alpha$, highlighting similar regions but with lower value compared to the other parameters.



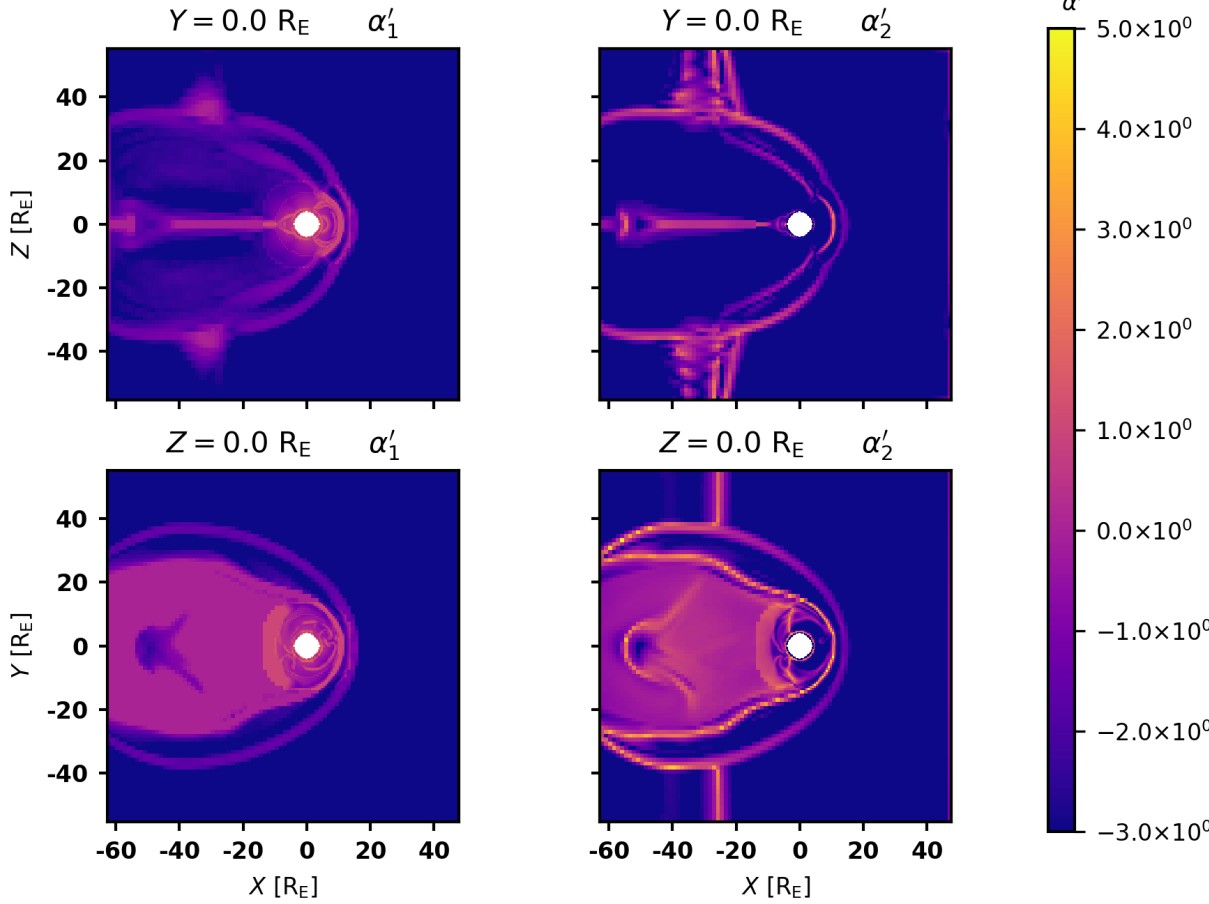

**Figure 4.** Plots of $\alpha'_1$ (left) and $\alpha'_2$ (right) on the $XZ$ (above) and $XY$-planes (below). Both indices highlight the magnetopause, particularly the subsolar region. Both indices also highlight the shock; $\alpha_1$ the subsolar shock better, and $\alpha_2$ the flanks and the foreshocks at $x \approx -30\,\mathrm{R_E}$.





## 2.5 Implementation

The refinement procedure is as follows:

1. Each process calculates the refinement indices for all its cells.

2. Each process iterates over its cells, marking them for refinement or coarsening based on the indices.

3. The requirement of neighboring cells having a maximum refinement level difference of one is imposed by dccrg, inducing additional refinement.

4. The memory usage after refinement is estimated before execution. If this exceeds the memory threshold set for the simulation, the run tries to rebalance the load according to the targeted refinement. If it is still estimated that the run will exceed the memory threshold, the run exits so the simulation may be restarted from that point with more resources or less aggressive refinement.

5. Refinement is executed. Refined cells are split into eight children with the distribution copied over, and coarsened cells are merged with the distribution averaged between eight siblings.

6. The moments are calculated for coarsened cells and coordinate data is set for all new cells.

7. The computational load is balanced between the processes and remote neighbor (ghost cell) information is updated.

Mesh refinement prefers refinement over coarsening. If either refinement index passes its refinement threshold, the cell is refined. A cell is coarsened only if both indices fir the cell and all its siblings pass their coarsening threshold. This overrides dccrg behavior for induced refinement; a cell will not be unrefined if that would result in the unrefinement of any cell above its coarsening threshold. Alternatively, either refinement index may be disabled in which case the other controls the refinement entirely. There is also additional bias against isolated, unrefined cells: a cell is always refined if a majority of its neighbors are refined, and cells are never unrefined in isolation; cells are only unrefined if they either have coarser neighbors or neighbors that would also unrefine.

Mesh refinement may be limited to a specific distance from the origin to limit refinement to where it is most relevant for the simulation; refinement is still induced outside this, as refining a cell on the boundary to the 2$^{\text{nd}}$ level requires cells outside to be on the first level and so on. Refinement may also be started at a specific time after the beginning of the simulation. This enables simulating initial conditions using a coarse grid and refining after a certain time without user intervention.

Boundary cells are not refined or unrefined dynamically, since this makes the refinement simpler and the code easier to maintain due to not having to factor in boundary conditions. Therefore it is recommended to initially refine the inner boundary region up to the desired level, similarly to static refinement.

Refinement may be done automatically during runtime or manually by creating a file named `DOMR` in the run directory. As refinement requires load balancing, it is done before load balancing at user-set intervals. Refinement cadence thus depends on load balance interval.





## 3 Results

### 3.1 Refinement

Runtime adaptive refinement was tested using similar parameters to a two level test run shown in Figure 2. Using only $\alpha_1$, three
220 test runs were done using refinement thresholds of $0.6$, $0.4$ and $0.2$ with the unrefinement threshold set to half the refinement
threshold. These test runs are referred to as alpha1-low, alpha1-med and alpha1-high respectively. Similarly, using only $\alpha_2$
three test runs were done using the same refinement and unrefinement thresholds. These are referred to as alpha2-low, alpha2-
med and alpha2-high. The option of delaying refinement was utilized to initialize the simulation with minimal refinement,
and refinement was enabled from $500\,\mathrm{s}$ onwards. Refinement was restricted to a radius of $50\,\mathrm{R_E}$ from the origin. The runs
225 alpha1-med and alpha2-med have final phase space cell counts closest to the control runs $1.058 \times 10^{11}$, yielding good points
of comparison.

A quantitative comparison of the runs is provided in Figure 5. Panels (a) and (b) show the behavior of phase-space cell count
on AMR runs. On minimal refinement the cell count and computational load is consistently lower than on the control run.
Notably the medium AMR runs with comparable cell count to control have more level 1 cells and less level 2 cells than the
230 control run; isolated regions of level 2 cells cause comparatively more induced refinement. The panels (c) and (d) showing the
spatial volume of each refinement level as a function of $\alpha_1$ are validation of $\alpha_1$-refinement, while the panels (e) and (f) show
the spatial volume of each refinement level as a function of $\alpha_2$ and likewise validate $\alpha_2$-refinement. The runs alpha1-med and
alpha2-med have few level 2 cells below the unrefinement threshold of $0.2$ compared to the control run, as well as having no
cells below level 2 above the refinement threshold of $0.4$. There are many level 1 cells below the unrefinement threshold, likely
235 due to induced refinement. The amount of cells above the refinement threshold is also reduced overall, as smaller cells end up
with lower cell-to-cell differences in variables compared to larger cells. Figures (g) through (j) show the same runs using the
$\alpha'$ indices or target refinement levels (4). As expected, there are sharp limits at $\alpha' = 0$ above which no cells below level 1, and
$\alpha' = 1$ above which there are no cells below level 2.

Examining plots of alpha1-med and alpha2-med in the $XY$ and $YZ$ planes (Figure 6), adaptive refinement follows the
240 structures of the magnetosphere better than static refinement. The spherical region around the inner boundary seen in Figure
2 is absent in both AMR runs, with refinement regions instead following the dayside magnetopause and shock. The tails of
refinement also flare wider in the $XY$-plane in both runs. In the case of $\alpha_1$, refinement has left some disjoint regions inside the
initial spherical refinement region. A smaller region of initial refinement might have avoided this issue. The results of the two
refinement methods are somewhat similar, but still distinct enough to warrant the usage of both.

245 In proton density two foreshocks in the $YZ$-plane can be seen, and adaptive refinement follows the moving structure. Com-
paring particle density to Figure 2 reveals the structure is somewhat different, particularly in the $XY$-plane; coarse initialization
changes the physical behavior of the system, with the benefit of smaller resource usage.

Refinement was done every load balance, i.e. every 50 timesteps. This corresponds to about half a second in simulation time.
Since cell size is between $2\,000$ and $8\,000\,\mathrm{km}$ in this simulation and solar wind plasma velocity is some hundreds of kilometers
250 per second, refinement should occur faster than the movement of any structure. The cadence seems sufficient: refinement





follows the foreshocks, and the general structure of refinement regions sets in quickly. Further testing is required to determine suitability in more dynamic conditions. These findings seem to suggest that this would only need an adjustment of cadence.







**Figure 5.** Ten plots of refinement-related parameters. Panel (a) shows the phase-space cell amount over simulation time for the static runs and the six AMR runs with the unrefinement and refinement threshold in brackets for $\alpha_1$ runs and the tuning parameter $a$ for $\alpha_2$ runs. Panel (b) shows the amount of phase-space cells in each refinement level at the end of the static runs and both med AMR runs ($t = 550\,\mathrm{s}$). The middle four panels show stacked histograms of spatial volume by $\alpha_1$ and $\alpha_2$ in the control and medium AMR runs with the lowest bin of $\alpha < 0.04$ clipped. The bottom four panels show stacked histograms of spatial volume by targeted refinement levels $\alpha_1'$ and $\alpha_2'$ in the control and medium AMR runs.

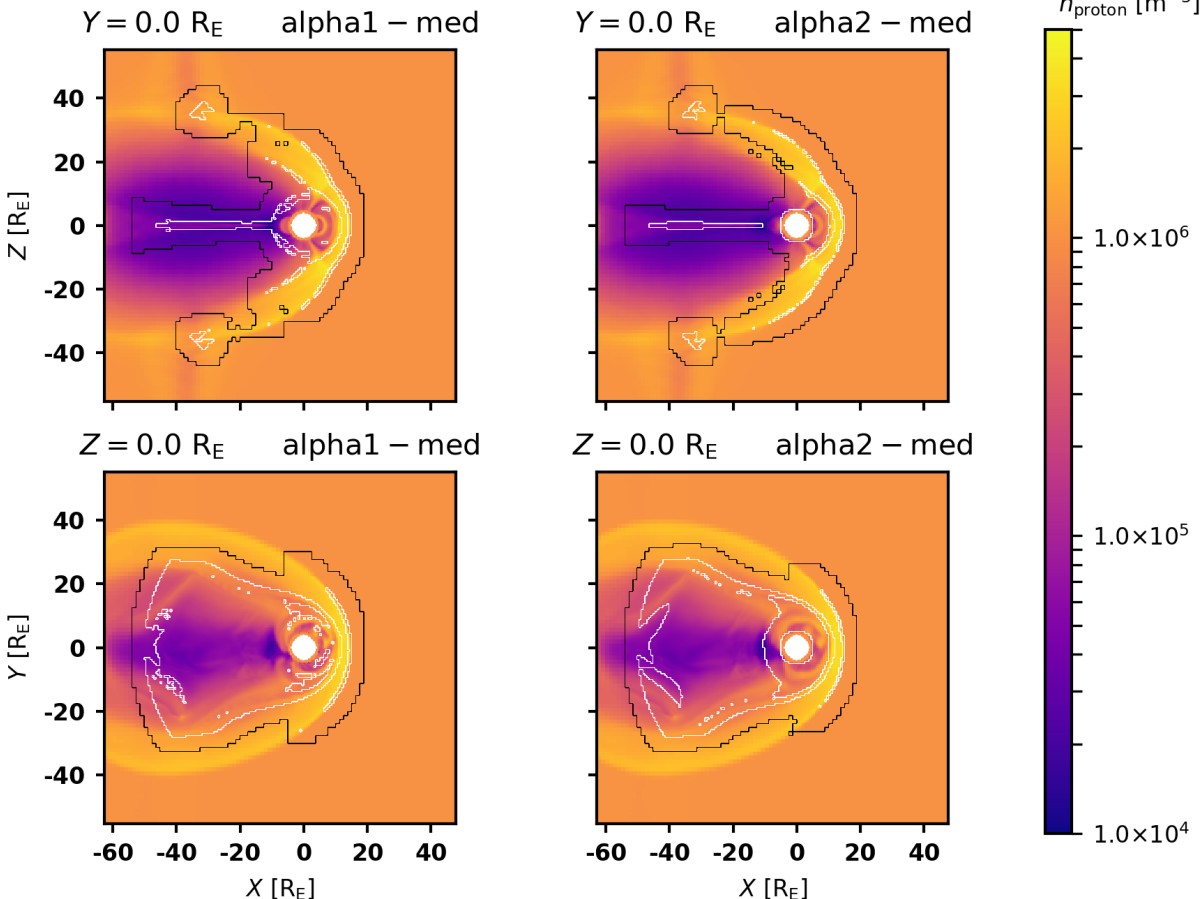

**Figure 6.** Contour plots of refinement level on top of particle density in the alpha1-med (left) and alpha2-med (right) runs, $t = 550\,\mathrm{s}$, top panels $XZ$-plane, bottom panels $XY$-plane. The run alpha1-med retains a large amount of the inner magnetosphere refined and has a clean level 2 boundary right on the bow shock. Meanwhile alpha2-med has less refinement in the inner magnetosphere, but has more on the sides of the shock. A faint shape can be seen in proton density around $x = -20$ to $-40\,R_\mathrm{E}$ in the top panels; this is a foreshock that the refinement picks up, which has partially exited the refinement radius.



## 3.2 Perfomance

Every run's performance was evaluated using the rate of cells solved per second per core. This data is provided in Table 1. The
AMR runs have generally worse performance relative to the control run, with the comparative cell-rate ranging from $62.6\%$
for alpha2-low to $99.9\%$ for alpha1-high. This implies that the increased grid complexity comes at a higher computational cost.
There are two possible explanations for performance improving with cell count. First, as the problem size compared to core
count increases, each process ends up spending more time in solving cells compared to communicating data with its neighbors,
resulting in more ideal parallelization. Second, more refinement leads to more unified regions with less interfaces between
unrefined and refined cells. This might be improved by using some heuristic to merge isolated regions of refinement, but it
remains to be seen whether any simplification of the grid would offset the cost of additional phase-space cells.

Table 2 compares time spent in spatial translation and MPI wait times in runs alpha1-med and control; most of the time
loss in translation is in waiting for other processes to finish translation in each direction. The load balancing method used was
Zoltan's recursive coordinate bisection (Devine et al., 2002), with velocity space cell count as the weight. This seems to work
poorly for a complex grid, either due to orthogonal cuts being non-ideal or the single weights not accounting for different
translation times in each direction. Another possibility is poor weight calculation: weights are calculated on the step before
load balancing, and the load balance after refinement uses the weight calculated for the parent not accounting for changing
refinement interfaces. Since refinement was performed every load balance in these tests, every load balance was thus sub-
optimal, at least until the grid had settled to a point where few cells were refined on each pass. It remains to be seen if this
problem persists on a production scale run with more tasks.

Of particular note is the performance of the initial 500 seconds of simulation, done on minimal refinement with only the
inner boundary region refined to level 2. The total phase space cell count reached at $500\,\mathrm{s}$ was $3.298 \times 10^{10}$, with roughly the
same amount of cells solved per core-second as the control run on $4\,096$ cores. Initializing the simulation on minimal static
refinement is thus quite efficient, using only a third of the resources of static refinement shown in Figure 2. This performance
benefit is expected to increase when refining a simulation up to level 3 or more.

Table 3 shows time taken in load balance and refining in each run. Balancing the load every 50 timesteps and refining
before each load balance, re-refining took on average $1.2\%$ of the simulation time in the run alpha1-med. In comparison, load
balancing took $0.67\%$, so refining almost triples the overhead compared to just rebalancing. However, load balancing itself
took $81\%$ more time on the alpha1-med run, indicating that the refined grid is harder to partition. Splitting a cell effectively
increases its load balance weight eightfold; if a task domain contains a large amount of refined cells this increases the amount
of communication required to balance the load. This effect isn't limited to tasks where cells are refined: as these tasks migrate
cells to neighboring tasks, they will also have to migrate cells to other tasks in order to balance load. Scaled to phase-space cell
count the effect on load balancing is smaller in the alpha1-high run, indicating similar reasons as the overhead in translation
performance. On the other hand, refinement time per cell grows with grid size; this is likely due to additional checks for induced
refinement in dccrg. As the total spatial cell count in the grid grows with refinement, each process has more cells, and each
refined cell has more neighbors to check for induced refinement.





| Run | Cores | Cells solved [1/s] | Cells solved per core [1/s] | % of Control |
|---|---|---|---|---|
| Control | 12 800 | $2.492 \times 10^{10}$ | $1.946 \times 10^{6}$ | 100.0 |
| Unrefined | 4 096 | $8.630 \times 10^{9}$ | $2.107 \times 10^{6}$ | 108.3 |
| alpha1-low | 12 800 | $1.674 \times 10^{10}$ | $1.308 \times 10^{6}$ | 67.2 |
| alpha1-med | 12 800 | $1.836 \times 10^{10}$ | $1.434 \times 10^{6}$ | 73.7 |
| alpha1-high | 12 800 | $2.488 \times 10^{10}$ | $1.944 \times 10^{6}$ | 99.9 |
| alpha2-low | 12 800 | $1.560 \times 10^{10}$ | $1.219 \times 10^{6}$ | 62.6 |
| alpha2-med | 12 800 | $1.586 \times 10^{10}$ | $1.239 \times 10^{6}$ | 63.7 |
| alpha2-high | 12 800 | $1.836 \times 10^{10}$ | $1.434 \times 10^{6}$ | 73.7 |

**Table 1.** Table of rates of phase-space cells solved per second and per core in each run over 500 to 550 s. Cell-rate is generally worse for AMR runs, but improves for runs with more refinement.

| Timer | Spatial translation | Pre-update barriers | | | |
|---|---|---|---|---|---|
| Run | Total [s] | z [s] | x [s] | y [s] | Total [s] |
| alpha1-med | 15 970.1 | 4 697.5 | 1 731.8 | 1 047.4 | 7 476.7 |
| Control | 10 460.0 | 1 287.7 | 1 104.5 | 837.2 | 3 229.4 |
| Difference | 5 510.1 | 3 409.8 | 627.3 | 210.2 | 4 247.3 |

**Table 2.** Comparison of time spent in spatial translation and specific timers within in alpha1-med and control runs. Vlasiator performs translation one dimension at a time, updating ghost cells in between. Pre-update barriers refers to the time spent by processes waiting for other processes to complete translation in order to update ghost cells. 77 % of the time difference between AMR and control runs in spatial translation is explained by these increased waiting times.





| Timer | Load Balance | | | Refine | | |
|---|---|---|---|---|---|---|
| Run | Time [s] | Time [%] | Time / cell [μs] | Time [s] | Time [%] | Time / cell [μs] |
| Control | 87.1 | 0.510 | 5.267 | – | – | – |
| Unrefined | 69.6 | 0.446 | 4.268 | – | – | – |
| alpha1-low | 100.0 | 0.576 | 9.257 | 158.7 | 0.914 | 14.70 |
| alpha1-med | 157.5 | 0.666 | 9.359 | 285.7 | 1.208 | 16.97 |
| alpha1-high | 237.7 | 0.671 | 7.643 | 681.6 | 1.923 | 21.92 |
| alpha2-low | 98.8 | 0.563 | 9.689 | 158.0 | 0.900 | 15.49 |
| alpha2-med | 129.9 | 0.577 | 9.003 | 219.4 | 0.974 | 15.21 |
| alpha2-high | 197.7 | 0.604 | 8.776 | 429.5 | 1.313 | 19.06 |

**Table 3.** Table of time taken in load balance and refinement in seconds, percentage of simulation time and microseconds per phase-space cell over 500 to 550 s. Load is balanced every 50 timesteps in each run, with the AMR runs refined before every load balance. Load balancing takes somewhat longer on AMR runs, with worst results on the runs with closest cell counts to the control.



## 4 Conclusions

In this paper we introduced a method to automatically adapt the Vlasiator spatial grid to concentrate numerical accuracy in regions of special interest. The method is based on two indices $\alpha_1$ and $\alpha_2$, measuring rate of change in spatial variables and the occurrence of current sheets respectively. The grid is refined to a higher resolution in regions where these indices are high, and coarsened to a lower resolution where they are low.

We also tested the performance of adaptive mesh refinement, and the results in Section 3 show this method works well for global simulations with some caveats. The option to delay refinement alone cuts computational load of the initialization phase to one third in the test setup as demonstrated in Table 1, and AMR produces good refinement albeit with notable performance overhead of around $26\%$ in the test case most similar to control. Since the performance difference between the control and AMR runs seems to be primarily caused by load imbalance, developing better load balance criteria and methods might help alleviate the issue.

Another possibility is to consider different refinement parameters. Replacing them in the simulation code is simple now that the groundwork for AMR has been laid out. As the current criteria borrow heavily from GUMICS and MHD-AEPIC, magnetohydrodynamic and embedded PIC simulations respectively, they might not be optimal for a Vlasov simulation. As both refinement indices are based on spatial variables, they do not explicitly account for kinetics present in the simulation. Implementic kinetic measures such as non-Maxwellianity (Graham et al., 2021) or agyrotropy (Swisdak, 2016) would efficiently indicate regions where kinetic phenomena dominate, but would not directly map to dimensionless gradients. Thus, refining those regions would not bring the evaluated parameter into the requested range and implementing them as refinement criteria is not straightforward.

Adaptive mesh refinement fulfils the goals set in its development: replacing static refinement with an adaptive and efficient algorithm. We plan to use AMR in upcoming large scale production runs, providing further information on the method's advantages and shortcomings. In particular, initializing a simulation at a low resolution allows for a longer total simulated time using the same amount of resources; however, care must be taken so this does not compromise the simulation results.



*Code and data availability.* The current version of model is available from Github: https://github.com/fmihpc/vlasiator/ under the GNU
General Public License Version 2 (GPLv2). The exact version of the model used to produce the results used in this paper is archived on
Zenodo (Pfau-Kempf et al., 2024), while the input data and scripts to run the model and analyze the output for all the simulations presented
in this paper are archived on Fairdata (Kotipalo, 2023). Figure 1 was done using VisIt 3.3.1 (Childs et al., 2012) and the rest using the
Analysator library available from Github: https://github.com/fmihpc/vlasiator/ under the GNU GPLv2 and archived on Zenodo (Battarbee
et al., 2021)

## Appendix A:  Reproducing the data

The data may be reproduced in the following manner using the provided configuration (Kotipalo, 2023):

1. Install and compile Vlasiator using the instructions provided in
   https://github.com/fmihpc/vlasiator/wiki/Installing-Vlasiator.

2. To generate the control data, run Vlasiator in the `control`-directory using the corresponding config via e.g.
   `srun BIN --run_config control.cfg` where `BIN` is the Vlasiator executable.

3. To generate the unrefined data, run Vlasiator in the unrefined-directory first using the configuration file `unrefined-500s`.
   Run again using the configuration file `unrefined-550s` using the restart file created at 500 simulation seconds via e.g.
   `srun BIN --run_config unrefined_550s.cfg --restart.filename restart/FILENAME`, where
   `FILENAME` is the restart file.

4. To generate each AMR data, run Vlasiator in each of the `alpha1` and `alpha2`-directories using the corresponding
   configuration files and the same restart file used for `unrefined-550s`.

5. The data in Tables 1, 2 and 3 is given by the script `data.sh`. Data for each run is given by `./data.sh RUN` where
   `RUN` is the path of the run to analyze.

6. The first three lines of output correspond to the first three columns of Table 1. The last column is the ratio of cells solved
   per core to the corresponding number of the control run.

7. The first four columns of Table 2 correspond to the timers `Spatial-space` and
   `barrier-trans-pre-update_remote{z,x,y}`. The average value in seconds is used here, with the total for
   Pre-update barriers calculated as the sum of the z, x, and y barriers and the difference is simply the difference between
   the two runs.

8. The columns Load Balance and Refine in Table 3 correspond to the timers `Balancing load` and
   `Re-refine spatial cell`. The values used are average time, percentage of time and time per phase space cell.



*Author contributions.* Leo Kotipalo prepared the manuscript with contributions from all co-authors, implemented adaptive mesh refinement in code and carried out the tests. Minna Palmroth wrote the abstract and is the University of Helsinki PI of Plasma-PEPSC.

*Competing interests.* The authors declare that they have no conflict of interest.

*Acknowledgements.* The paper is based on code development as part of the Plasma-PEPSC project, grant number 4100455. Markus Battarbee is funded by the Academy of Finland grant 335554. Yann Pfau-Kempf is funded by the Academy of Finland grant 339756. Minna Palmroth is funded by the Academy of Finland grants 336805, 345701 and 347795. The simulations and data analysis presented was done on the CSC Mahti supercomputer.

The Finnish Centre of Excellence in Research of Sustainable Space, funded through the Academy of Finland grants 312351 and 1336805, has significantly supported Vlasiator development, as has the European Research Council Consolidator Grant 682068-PRESTISSIMO. The authors wish to thank the University of Helsinki local computing infrastructure and the Finnish Grid and Cloud Infrastructure (FGCI) for supporting this project with computational and data storage resources.



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
