# Peer review of "Physics-motivated Cell-octree Adaptive Mesh Refinement in the Vlasiator 5.3 Global Hybrid-Vlasov Code"

_EGUsphere, 2024_

## Author Response (AR1)

We thank the referees for their comments and feedback, addressed below.

**RC1**

**Line 64: Does Vlasiator use the real mass ratio?**

Vlasiator uses the real mass ratio for protons and electrons since in the hybrid model electrons aren't explicitly modelled, obviating the need for a non-physical mass ratio. Real electron masses are used in the electron pressure gradient term of Ohm's law.

   Revision: Clarified usage of physical mass ratio in Section 1.1.

**70: Please elaborate more about what spatial optimizations are needed.**

The further spatial optimizations on line 70 are in reference to mesh refinement introduced in the following sub-section. We will clarify this in the revision.

   Revision: Moved and edited this sentence from end of Section 1.1 to start of Section 1.2.

**Line 84: AMR is not used in MHD-AEPIC. MHD-AEPIC is dynamically apply kinetic physics in regions that are needed. The grid resolution is not varying in space. BATS-R-US indeed uses AMR in the MHD grid. I suggest removing the MHD-AEPIC citation here.**

MHD-AEPIC citation will be removed from line 84, thank you for the correction

   Revision: Removed citation.

**Figure 5(a): Why the cell number increases for the unrefined run?**

In figures 5a and 5b the plotted cell count is the entire phase-space i.e. including velocity space which fills out as the state evolves. It is a measure of total computational/memory load. We will amend the figure to clarify this.

   Revision: Figure subtitles amended to phase-space cells for (a) and (b) and spatial volume for (c) through (j).

**Table 1: Please list the total runtime of each run in the table, that would justify using AMR in the future production runs, even the scaling seems to be bad.**

We will provide total runtime in Table 1.

   Revision: Total runtime provided in Table 1.

**RC2**

**Was OpenMP used in the test runs? Regardless of the answer, could the authors provide additional discussion on its impact on performance, particularly in terms of load balancing?**

OpenMP was used in all tests. Each process ran on two physical cores with hyperthreading, for a total of four OpenMP threads per process. Task balance was not the principal factor investigated, so every run used the same amount of cores per task. We will add the used threading settings in the revision.

Revision: Discussed task balance in Section 3.2.

**It is speculated that the load balance was non-ideal because the weights used in the load balancing algorithm did not account for changes in the mesh. Would more frequent load balancing help mitigate this issue?**

More frequent load balancing is expected to help with AMR performance, or rather, not refining before every load balance. The intention was to "stress-test" AMR to see the overhead of refinement itself on maximal cadence in comparison to load balance. Load balancing itself also incurs a cost, so too frequent load balances will also be inefficient.

Revision: Clarified that load balancing between AMR passes might help in Section 3.2.

**I noticed in the Acknowledgements that the simulations were performed on CSC Mahti supercomputer. However, while reading the manuscript, I had questions about this. It would help the reader if a brief description of the computer were included alongside the presentation of the test results**

Detailed information of the Mahti supercomputer is available in the Mahti documentation (https://docs.csc.fi/computing/systems-mahti/) – in these tests the CPU nodes were used. We will include a description in the revision.

Revision: Added description of the Mahti supercomputer in Section 3.2.

**Line 138, starting with 'These gradients are': Technically speaking, an expression like $(\Delta B)^2/(2\mu_0)$ is not the gradient of the magnetic field energy, which would be $B\Delta B/\mu_0$. To avoid confusion, could the authors revise this sentence?**

**There is a typo in Line 173. 'his' $\rightarrow$ 'this'?**

Thank you for the observant correction. The mention of gradients and the typo will be corrected in the revision.

Revision: Sentence revised, typo corrected.

**RC3**

**Line 52 - what is the ion kinetic scale compared to the resolution?**

The thermal Larmor radius ($\sim 130\,\text{km}$) or ion inertial length ($\sim 230\,\text{km}$) are not resolved in these runs, where the resolution is $2000\,\text{km}$ on the finest level. As also discussed by Ganse et al. (2023), even $1000\,\text{km}$ currently achieved at production scale is a compromise. Performing several case studies at production run resolution would require significant computational resources through one or more dedicated tier-0 resource applications. We will include this discussion point in the paper.

Revision: Added kinetic scales and discussion of resolution to Section 1.1.

**What are the physical parameters of the simulation being used for the study (eg plasma and field parameters)**

Thank you for pointing out this omission. The simulation was initialised with a solar wind proton temperature of $5 \times 10^5\,\text{K}$, proton number density of $1\,\text{cm}^{-3}$, and a solar wind velocity of $-750\,\text{km s}^{-1}$ in the x-direction. Within $30 \times 10^3\,\text{km}$ from the origin, the plasma is at rest, with velocity tapering sinusoidally from a distance of $30 \times 10^3\,\text{km}$ to a distance of $100 \times 10^3\,\text{km}$, from where the full solar wind velocity is used. The background field consists of $-5\,\text{nT}$ IMF flux in the $z$-direction and a vector dipole field equivalent to Earth's. We will provide the physical setup in the revision.

Revision: Added physical parameters in Section 1.1, noting in 3.1 that these parameters were used for the test runs.

**Figure 1 - What are the units in the colour plot? Also, the physics appears different in the two runs (the most obvious difference is the flux rope in the tail)?**

Units in Figure 1 are $1\,\text{m}^{-3}$, we will add this to the revision. Physical differences stem from the lower run only having run for $50\,\text{s}$ with AMR, with kinetic effects remaining unresolved during the $500\,\text{s}$ initialization on the lower resolution. Thus, the unresolved 500 second initialization period allows global but not kinetic features to form, and the full resolution should be allowed to impact the results for several ion gyroperiods before using the data for scientific analysis.

Revision: Added units to Figure 1 and discussion of physical differences in the caption.

**Line 239 - Should this be XY and XZ planes?**

Yes, we will correct this in the revision.

Revision: Corrected.

**Line 245 - What do the authors mean when referring to "foreshocks"? These do not appear to be the perturbations upstream of the quasi-parallel shock, which is the usual definition.**

We acknowledge that it is unusual to refer to the structures in Figure 6 as foreshocks. However, with the Bz IMF, the flanks do in fact build up to a quasi-parallel shock, even though the shock-normal plasma flow is miniscule. From a point of view of particle shock physics, we suggest that this region is still best described as a foreshock, even if the upstream plasma flows more in parallel with the shock front than directed towards it.

Revision: Clarified these as flank foreshocks.

**Figure 5 - The bar charts in (c)-(j) have some quantities obscured due to the overlapping. I would suggest modifying the opacity or a line plot.**

The histograms in Figure 5 (c)-(j) are stacked, i.e. the total height of the bars show the portion of spatial volume with some value of alpha, with the heights of the coloured sections of the bars showing the portion of cells on each refinement level. We apologise for the confusion and will clarify this in the revision.

Revision: Clarified that Figures 5 (c) - (j) are stacked histograms in the text.

**Figure 6 - It would be helpful to provide a plot of the control in addition to the existing plots**

I Thank you for the suggestion. Plots of the control run are given in Figure 2, and we will replicate them in Figure 6 for easier comparison.

Revision: Control run replicated in Figure 6.

**In general, the authors do not show that their simulations converge. This is alluded to in point 2), where the behaviour of the magnetotail looks different in the two runs. Without this information, whether the method works cannot be judged.**

Thank you for raising an interesting point. The physics observed in the control and AMR runs are different, as for example the lower run in Figure 1 has only been refined in the tail for 50 seconds. Indeed the simulations haven't converged and in an absolute sense cannot converge – different local resolutions affect physical phenomena such as reconnection rate through numerical resistivity (Rembiasz et al., 2017). It is also known that some kinetic phenomena cannot be described unless sufficient resolution is achieved (Dubart et al., 2020). Running longer with AMR enabled is expected to result in more qualitatively similar structures; the short 50 s period of refinement was meant to determine the shape of refined regions, thus ascertaining the relevance of the chosen refinement parameters.

Revision: Effects of resolution discussed in Section 1. Explanation of physical differences added in Section 3.1.

**Table 3 - I suggest providing the total run time in the table as well, since that is of practical use.**

We will provide this in Table 3.

Revision: Contrary to the comment and response, this was added to Table 1 as requested in RC1.

**Line 179 - Please define induced refinement.**

Thank you for the clarification request. Induced refinement is dccrg behaviour for enforcing maximum refinement differences between neighbours. Essentially, refining a cell whose neighbours are too coarse refines those neighbours. We will include this additional information in the revision.

Revision: Induced refinement explained in Section 2.1.

**I would suggest using a different font/italics for dccrg**

The authors acknowledge that the lowercase initialism dccrg is less than ideal for readability. We will capitalise it in the revision.

Revision: All mentions of DCCRG capitalised.

**Line 138 - move (a-e) to before the quantities. E.g (a) particle density ...**

**Line 173 - his → this**

**There are certain parts of the text where contractions are used (e.g don't). These do not fit stylistically with the rest of the paper.**

We will make these technical corrections in the revision.

Revision: Corrected.

**References**

Dubart, M., Ganse, U., Osmane, A., Johlander, A., Battarbee, M., Grandin, M., Pfau-Kempf, Y., Turc, L., and Palmroth, M. (2020). Resolution dependence of magnetosheath waves in global hybrid-vlasov simulations. *Annales Geophysicae*, 38(6):1283–1298.

Ganse, U., Koskela, T., Battarbee, M., Pfau-Kempf, Y., Papadakis, K., Alho, M., Bussov, M., Cozzani, G., Dubart, M., George, H., Gordeev, E., Grandin, M., Horaites, K., Suni, J., Tarvus, V., Kebede, F. T., Turc, L., Zhou, H., and Palmroth, M. (2023). Enabling technology for global 3D + 3V hybrid-Vlasov simulations of near-Earth space. *Physics of Plasmas*, 30(4):042902.

Rembiasz, T., Obergaulinger, M., Cerdá-Durán, P., Ángel Aloy, M., and Müller, E. (2017). On the measurements of numerical viscosity and resistivity in eulerian mhd codes. *The Astrophysical Journal Supplement Series*, 230(2):18.